# Gene Polymorphisms of NOD2, IL23R, PTPN2 and ATG16L1 in Patients with Crohn’s Disease: On the Way to Personalized Medicine?

**DOI:** 10.3390/genes12060866

**Published:** 2021-06-05

**Authors:** Peter Hoffmann, David Lamerz, Petra Hill, Marietta Kirchner, Annika Gauss

**Affiliations:** 1Department of Gastroenterology and Hepatology, University Hospital Heidelberg, INF 410, 69120 Heidelberg, Germany; Peterhoffmann84@gmail.com (P.H.); david.lamerz@web.de (D.L.); petra.hill@med.uni-heidelberg.de (P.H.); 2Department of Medical Biometry, Institute of Medical Biometry and Informatics, University Hospital Heidelberg, INF 130.3, 69120 Heidelberg, Germany; kirchner@imbi.uni-heidelberg.de

**Keywords:** inflammatory bowel disease, Crohn’s disease, NOD2, PTPN2, rs7234029, ATG16L1, IL23R, ustekinumab, anti-interleukin-12/23

## Abstract

Genetic and environmental factors are involved in the pathogenesis of inflammatory bowel diseases (IBD). The study aimed at investigating the potential influence of single nucleotide polymorphisms (SNPs) NOD2 rs2066844, NOD2 rs2066845, NOD2 rs2066847, IL23R rs11209026, PTPN2 rs2542151, PTPN2 rs7234029, and ATG16L1 rs2241880 on the response to immunomodulatory therapies and disease course in Crohn’s disease (CD). This is an uncontrolled retrospective monocentric study including patients from the IBD outpatient clinic of Heidelberg University Hospital. Therapy responses and disease courses were related to genetic findings. 379 patients with CD were included. The presence of at least one PTPN2 rs7234029 risk allele was associated with nonresponse to anti-interleukin-12/23 treatment (89.9% vs. 67.6%, *p* = 0.005). The NOD2 rs2066844 risk allele was associated with a first-degree family history of colon cancer (12.7% vs. 4.7%, *p* = 0.02), the ATG16L1 rs2241880 risk allele with ileal CD manifestation (*p* = 0.027), and the IL23R rs11209026 risk allele with a higher rate of CD-related surgeries per disease year (0.08 vs. 0.02, *p* = 0.025). The results of this study underline the relevance of genetic influences in CD. The association of the PTPN2 rs7234029 risk allele with nonresponse to anti-interleukin-12/23 treatment in CD patients is a novel finding and requires further investigation.

## 1. Introduction

Inflammatory bowel diseases (IBDs) are complex, poorly understood, incurable inflammatory diseases of the gastrointestinal tract with mostly relapsing or chronic disease courses. At present, more than one million residents of the USA and 2.5 million residents of Europe are estimated to suffer from IBD [1]. The main disease entities are Crohn’s disease (CD) and ulcerative colitis (UC). As the phenotypes and courses of these debilitating conditions are very heterogeneous, and as—so far—no ‘one size fits all’ therapy has been established for IBDs, there is a great medical need to identify biomarkers for disease behavior, prognosis and response to the growing spectrum of available medical therapies.

Both genetic and environmental factors are involved in the pathogenesis of IBD [2]. IBDs are polygenetic diseases with a prevalence of familial clustering in 5–10% of cases [3]. In genome-wide association studies of cases and controls, more than 200 IBD susceptibility genes have been reported [4].

NOD2 (nucleotide-binding oligomerization domain-containing protein 2) is located on chromosome 16q12.1 and was the first disease-susceptibility gene discovered for CD. NOD2 is a pattern recognition receptor that is involved in the homeostasis of intestinal immunity [5]. In response to bacterial infection, CARD15/NOD2 acts as an intracellular bacterial receptor and activates the kappa B nuclear factor (NF-κB) [6]. NOD2 mutations lead to dysregulation of host–microbe interactions, promoting inflammation in the ileum, which is characteristic of CD [7]. So far, more than 60 polymorphisms of this gene have been identified; however, three common mutations (Leu1007fsinsC, Arg702Trp, and Gly908Arg) have been specifically associated with ileal involvement, stricturing complications, and earlier disease onset [8].

Several studies have revealed associations between mutations in the NOD2 gene and low anti-TNFα through levels in CD patients [9,10].

Autophagy is involved in the pathogenesis of IBD in multiple ways, including secretion of antimicrobial substances from Paneth cells, clearance of invading pathogens, presentation of antigens, and production of proinflammatory cytokines by macrophages [11]. Single nucleotide polymorphisms (SNPs) of autophagy genes such as autophagy-related gene 16 like 1 (ATG16L1) have been linked to CD. ATG16L1 affects cellular autophagy processes and bacterial clearance in immune cells and may affect the gut microbiota in IBD patients [12]. The most common and extensively studied genetic variant of ATG16L1 (rs2241880; leading to a T300A conversion) exhibits a strong association with the risk of developing CD. It plays a crucial role in pathogen clearance, resulting in imbalanced cytokine production, and is linked to other biological processes, such as the endoplasmic reticulum stress/unfolded protein response [13].

The gene encoding PTPN2 has reached great clinical significance due to the association of several SNPs with its locus (18p11). These SNPs have been shown to be involved in chronic inflammatory conditions as CD, UC, type 1 diabetes, and celiac disease [14,15,16]. The rs2542151 SNP is the most widely identified and most thoroughly analyzed PTPN2 SNP associated with IBD [17]. The rs7234029 SNP may be associated with a stricturing disease phenotype in CD patients and early disease onset in both CD and UC patients [18].

Finally, genome-wide association studies have identified an association of the proinflammatory cytokine interleukin-23 receptor subunit (IL23R) with CD and UC. A meta-analysis considering 41 publications with 13803 CD patients and 5876 UC patients revealed that the IL23R rs11209026 polymorphism significantly decreased the risk of developing CD or UC. A subgroup analysis based on ethnicity showed that the IL23R rs11209026 SNP was associated with the risk of developing UC in Caucasians, but not in Asians (*p*  >  0.05) [19].

With the cited facts in mind, the goal of the present study was to assess the impact of several promising SNPs on the response of CD activity to available IBD-directed drugs, as well as on CD disease courses and prognosis.

## 2. Materials and Methods

This is an uncontrolled monocentric retrospective observational study including outpatients with moderate to severe CD at a German university hospital serving as a tertiary referral center for the treatment of IBD patients.

All CD patients who visited the IBD outpatient clinic of Heidelberg University Hospital between 1 May 2017 and 1 June 2020, and who had consented to contribute a blood sample to the local IBD biobank, were included in the study. Demographic and clinical data were collected from fully electronic patient records. The follow-up ended for all patients on 23 January 2021.

The study was approved by the local Ethics Committee (Alte Glockengiesserei 11/1, 69115 Heidelberg; protocol number: S-297/2020). Written informed consent was required for participation in the IBD biobank (protocol number: S-238/2017).

The primary study objective was to investigate potential associations of the SNPs NOD2 rs2066844, NOD2 rs2066845, NOD2 rs2066847, IL23R rs11209026, PTPN2 rs2542151, PTPN2 rs7234029, and ATG16L1 rs2241880 with therapy outcomes in our cohort of CD patients.

Secondary study objectives were to investigate potential associations of the named SNPs with baseline characteristics.

Inclusion criteria were: age ≥ 18 years, ascertained diagnosis of moderately to severely active CD according to ECCO criteria [20], informed consent to participate in the local IBD biobank, and donating a blood sample. Exclusion criteria were: diagnosis of indeterminate colitis or UC, age < 18 years, denial to take part in the IBD biobank sampling.

Information retrieved from electronic patient charts included gender, age at first diagnosis of CD, disease duration at time of observation, disease phenotype and disease behavior according to Montreal classification [21], family history of IBD and colon cancer, smoking status, presence of extraintestinal manifestations, history of IBD-related intestinal surgery, medical IBD therapies (anti-TNFα, anti-integrin, anti-interleukin 12/23, JAK (Janus kinase)-inhibitor, or immunomodulator treatment with azathioprine, 6-mercaptopurine, methotrexate), and the number of different prior medical IBD therapies.

Genomic deoxyribonucleic acid (DNA) was isolated from EDTA-anticoagulated peripheral venous blood using a QIAamp DNA Blood Midi kit according to the manufacturer’s protocol (Qiagen GmbH, Hilden, Germany).

The blood samples from CD patients were analyzed for (1) the three main CD-associated NOD2 SNPs rs2066844, rs2066845 and rs2066847; (2) the IL23R SNP rs11209026; (3) the two PTPN2 SNPs rs2542151 and rs7234029; and (4) the ATG16L1 SNP rs2241880.

The genotypes of all named mutations were screened by a LightCycler^®^ 480 Instrument I (Roche Diagnostics International AG, Rotkreuz, Switzerland). Polymerase chain reaction (PCR) was performed with the LightCycler^®^ fast start DNA Master HybProbe according to the manufacturer’s protocol (Roche Diagnostics International AG, Rotkreuz, Switzerland). PCR was performed as follows: denaturation at 95 °C for 10 min, followed by 45 cycles of 10 sec at 95 °C, 10 sec at 60 °C, and 15 sec at 72 °C. The primers were designed and synthesized by TIB MOLBIOL GmbH (Berlin, Germany) according to the dbSNP database of NCBI (https://www.ncbi.nlm.nih.gov/projects/SNP, accessed on 10 May 2021). Primers and PCR conditions are specified in Table 1.

Disease extent and disease behavior were categorized according to the Montreal classification for CD [21]. Nonresponse to medical therapy was defined by a switch to a different medical therapy within six months of treatment initiation due to primary lack or secondary loss of efficacy or need for surgery. Response to medical therapy was defined by the patient remaining on the same medical therapy through month 6 from treatment initiation.

Descriptive statistics were calculated as absolute (*n*) and relative (%) frequencies for categorical variables and as medians with interquartile ranges (IQR) or means with standard deviations (SD) for continuous variables dependent on presence of normal distribution. The primary outcome ‘response to therapy’ was considered for every therapy separately and restricted to the patients who were exposed to the respective therapy for at least six months, or showed nonresponse within the first six months. The respective sample size is indicated for every therapy. Therapies with a duration of <6 months for other reasons, such as adverse events, infections, incompliance, loss to follow-up or insufficiently documented reasons, were excluded in the statistical analysis of therapy responses, but the frequencies of these events are presented.

To identify potential associations of every variable with SNP mutations (wild type vs. risk type), the Mann–Whitney U test was used for continuous variables, and Chi-squared tests or Fisher’s exact test in case of small expected frequencies (*n* < 5) for categorical variables. This is an exploratory analysis presenting descriptive *p*-values not adjusted for multiple testing. A *p*-value < 0.05 was considered statistically significant. The statistical analyses were performed using SPSS Statistics 24 (IBM, Chicago, IL, USA).

## 3. Results

### 3.1. Characteristics of the Included Patients

Three hundred seventy-nine CD patients met the inclusion criteria and were included in the study (Table 2). Among them, 44.6% were male, and the median age at first diagnosis was 24 years. The disease location was ileal in 31.1%, colonic in 11.5%, and ileocolonic in 57.4%. Upper gastrointestinal involvement was noted in 12.6% of the patients. Disease behavior was non-stricturing and non-penetrating in 36.9%, stricturing in 24.3%, and penetrating in 38.8% of the patients. Perianal disease was diagnosed in 28.4% of the patients.

A positive family history of IBD in first-degree relatives was found in 16.6% and a positive family history of colon cancer in first-degree relatives in 6.5% of the patients. At least one extraintestinal manifestation was documented in 54.1% of the patients, and prior IBD-related surgery had been performed in 53%. Previous medical therapies included immunomodulatory therapies with azathioprine and/or methotrexate in 91.6% of the patients, anti-TNFα therapy in 80.7%, anti-integrin (vedolizumab) therapy in 21.7% and anti-interleukin-12/23 (ustekinumab) therapy in 34.3%. Only 8.4% of the included CD patients were naive to immunosuppressive therapy. Table 3 presents the results of PCR analyses and distribution of SNPs in CD patients.

### 3.2. Associations of SNPs with Therapy Responses

NOD2 rs2066844, NOD2 rs2066847, ATG16L1 rs2241880, IL23R rs11209026 and PTPN2 rs2542151 risk alleles were not associated with therapy responses in our CD patients. The NOD2 rs2066845 risk allele was linked to a greater total number of prior IBD-directed medical therapies (*p* = 0.023), which was not confirmed if the number of therapies per disease year was considered. However, the SNP was not found to be associated with therapy response.

Most interestingly, the PTPN2 rs7234029 risk allele was clearly associated with nonresponse to anti-interleukin-12/23 treatment (89.9% vs. 67.6%, *p* = 0.005). As shown in Table 4, no relevant difference in the baseline characteristics between subgroups was found which could have explained this association.

### 3.3. Secondary Study Objectives

The NOD2 rs2066844 risk allele was associated with a first-degree family history of colon cancer (12.7% vs. 4.7%, *p* = 0.02). In addition, we found a near-significant association of the NOD2 rs2066844 risk allele with a history of IBD-related surgery (65.2% vs. 52.8%, *p* = 0.064; or, if corrected to the number of operations per disease year, 0.05 vs. 0.03, *p* = 0.091). The NOD2 rs2066845 risk allele showed a tendency to be more prevalent in patients reporting first-degree relatives with IBD (26.8% vs. 15.3%, *p* = 0.065), as well as in earlier disease onset according to the Montreal classification (*p* = 0.068). Finally, the NOD2 rs2066847 risk allele was associated with younger age according to the Montreal classification (*p* = 0.035) and was more prevalent in patients with the ileal phenotype according to the Montreal classification (*p* = 0.088) and in patients having first-degree relatives with IBD (23.7% vs. 15.2%, *p* = 0.113).

The ATG16L1 rs2241880 risk allele was associated with ileal CD manifestation according to the Montreal classification (*p* = 0.027). Concerning the IL23R gene, its rs11209026 risk allele was linked to a higher rate of CD-related surgeries per disease year, indicating a more complicated disease course (0.08 vs. 0.02, *p* = 0.025).

The PTPN2 rs2542151 risk allele showed no association with disease characteristics, whereas the PTPN2 rs72234029 risk allele was associated with a smaller number of CD-related surgeries per disease year (0.000 vs. 0.037, *p* = 0.048).

Results of primary and secondary study objectives are shown in Appendix A.

## 4. Discussion

In the light of an urgent medical need for personalized therapeutic strategies in IBD, the present study had the goal of evaluating polymorphisms of known IBD-related genes with regard to potential associations with disease phenotypes and therapeutic responses in CD patients. The analyzed cohort of our tertiary IBD care center is characterized by relatively severe and refractory disease courses with a history of CD-related surgical interventions in 53% and a history of immunomodulatory treatment and anti-TNFα treatment in 91.6 and 80.7% of the included CD patients, respectively.

The genetic background of CD becomes obvious in our patient cohort, where 16.6% of the patients indicated having at least one first-degree relative who also suffered from IBD. A review focusing on familial risk in IBD reported prevalences of IBD in first-degree relatives of 0.35–4.5% in CD patients, and 0.3–2.7% in UC patients [22]. Up to 12% of all IBD cases were found to be clustered within families [23].

One of the key results of our study is that SNPs of the IBD-related genes of interest were present in up to one third of our cohort, if both the heterozygous and homozygous variants were considered. In our CD patients, the prevalences of homozygous or heterozygous NOD2 rs2066844, NOD2 rs2066845, NOD2 rs2066847, ATG16L1 rs2241880, IL23 rs11209026, PTPN2 rs2542151 and PTPN2 rs7234029 polymorphisms were 20.4%, 12.5%, 18.1%, 82.3%, 6.6%, 30.3% and 30.8%.

The only SNP which did not occur homozygously was the IL23R SNP rs11209026. The rarity of this variant in our study is in line with a Spanish study which found GG:GA:AA prevalences of 83.8%:15.9%:0.3% in 598 CD patients [24].

The risk allele frequencies of NOD2 rs2066844, NOD2 rs2066845, NOD2 rs2066847, ATG16L1 rs2241880, PTPN2 rs2542151 and PTPN2 rs7234029 were similar to the ones described in other studies [18,25,26].

Most interestingly, we found an association of the PTPN2 rs7234029 risk allele in CD patients with nonresponse to anti-interleukin-12/23 treatment, considering a total of 110 CD patients who were treated with an interleukin-12/23-inhibitor. Is there a possible functional explanation for this finding? An in silico analysis revealed that PTPN2 rs7234029 potentially modulates the binding sites of several transcription factors involved in inflammatory processes, including GATA-3, NF-κB, C/EBP and E4BP4 [18]. This may be a possible link to the nonresponse to anti-interleukin-12/23 therapy that we observed in our study. Due to the association we found in our study, the role of PTPN2 rs7234029 as a potential biomarker for the success of ustekinumab therapy in CD patients should be further investigated in larger prospective cohorts of CD patients.

Among our CD patients, carriers of the NOD2 SNP rs2066844 had significantly more often first-degree relatives with colon cancer than non-carriers. A meta-analysis of 30 studies in Caucasians revealed a higher risk for colorectal cancer in patients with NOD2 SNPs rs2066844, rs2066845 and rs2066847 [27]. Most likely due to the overall rare occurrence of colorectal cancer, we could only find this association for NOD2 rs2066844, but not for NOD2 rs2066845 or NOD2 rs2066847. The NOD2 rs2066847 risk allele was associated with younger age at diagnosis according to the Montreal classification and with fewer prior therapies. In contrast to the NOD2 SNP rs2066845, this might imply that the therapies were more effective in patients with the risk allele of NOD2 rs2066847.

The tendency of an association with the ileal phenotype in NOD2 rs2066847 carriers is in accordance with the results from Schnitzler et al. [26], who described that all homozygous carriers of this mutation in 1.076 CD patients presented with ileal CD, and if they were smokers, they developed ileal stenosis in all cases.

In addition, the ATG16L1 polymorphism rs22041880 was associated with the ileal CD phenotype (OR: 1.786; CI 0.940–3.392). In accordance with this finding, a 2.2-fold risk for ileal disease was detected in a UK cohort of CD patients [27].

The presence of at least one IL23R rs11209026 risk allele was associated with a greater number of IBD-related surgeries per disease year in our CD patients (*p* = 0.025). In contrast, another group found a marginally lower risk for surgical interventions in CD patients carrying the IL23R rs11209026 risk allele (*p* = 0.067) [28].

One clear strength of this study is the overall distinctive genetic association of IBD in our cohort, which can be presumed from the high rate of first-degree relatives with IBD in our cohort. Another strength of the study is the homogeneity of the cohort of relatively treatment-refractory patients, who are typically treated at IBD outpatient clinics of German university hospitals. As the patient interviews at our clinic have been performed according to a standard operating procedure since 2017, sufficient structured information was available about the patients‘ backgrounds and disease courses.

A limitation is that no control group was included. However, all of the analyzed SNPs are already known to be associated with IBD in Caucasians, so the main focus of the study was to gain a deeper insight into potential associations with disease courses and—especially—responses to common IBD therapies. Steroid use was not considered as many patients handle steroids on their own. In a retrospective study, there is no possibility to reliably measure cumulative steroid doses. Thus we decided to choose a longer time frame than commonly used to assess therapy response in our study, because the type of immunosuppressive therapy is usually changed in patients who are not able to taper off their steroids at the most until month 6 after the initiation of a new immunosuppressive medication. Other limitations of the study are that subgroups were too small for multivariable analyses, and that different effects concerning the functionality of homozygous versus heterozygous genotypes were not considered separately due to the overall small cohort size.

## 5. Conclusions

The findings of this study underline the relevance of genetic influences in CD and suggest the exploration of larger patient cohorts in order to identify genetic biomarkers related to therapy response. In particular, the novel finding of an association of the PTPN2 rs7234029 risk allele with nonresponse to anti-IL-12/23 treatment in CD patients calls for further investigation, as there is an urgent need for biomarkers to predict therapy responses in IBD patients. Further studies exploring functional effects of PTPN2 rs7234029 are needed.

## Figures and Tables

**Table 1 genes-12-00866-t001:** Primers for polymerase chain reaction.

Wild Type Gene	Primers, 5’-3’
NOD2 R702W	CCAGACATCTGAGAAGGCCCTGCTC
rs2066844	GGCGCCAGGCCTGTGCCCGCTGGTG
NOD2 G908R	CTCTTTTGGCCTTTTCAGATTCTGG
rs2066845	GCAACAGAGTGGGTGACGAGGGGGC
NOD2 3020insC	GGGGCAGAAGCCCTCCTGCAGGCCC
rs2066847	TTGAAAGGAATGACACCATCCTGGA
IL23R R381Q	GATTGGGATATTTAACAGATCATTCC
rs11209026	AACTGGGTAGGTTTTTGCAGAATTT
PTPN2	ACTTCGCCAATGCCTTGGTTCGGGC
rs2542151	CTTCCTGAGACTCTCATTTTCCTAA
PTPN2	ACACTAGCAGATATTGTAACATCAG
rs7234029	TAAGTCACAACACTGTATTGGCCCA
ATG16L1 T300A	ACTTCTTTACCAGAACCAGGATGAG
rs2241880	ATCCACATTGTCCTGGGGGACTGGG

ATG16L1: autophagy related 16 like 1; IL23R: interleukin-23 receptor; NOD2: nucleotide-binding oligomerization domain 2; PTPN2: Protein tyrosine phosphatase non-receptor type 2.

**Table 2 genes-12-00866-t002:** Baseline characteristics and therapy outcomes of the included patients (*n* = 379).

Variable	CD Patients
*n*	379
Male, *n* (%)	169 (44.6)
Disease duration at baseline (years), mean ± SD	17.4 ± 11.5
Age at diagnosis (years), mean ± SD	27.6 ± 11.9
Montreal classification of CD:	
Age, *n* (A1:A2:A3)	36:285:58
Phenotype, *n* (L1:L2:L3), *n* = 373	116:43:214
Phenotype L4, *n* (%), *n* = 373	47 (12.6)
Behavior, *n* (B1:B2:B3), *n* = 366	135:89:142
Perianal disease, *n* (%), *n* = 366	104 (28.4)
First-degree relative(s) with IBD, *n* (%)	60 (16.6), *n* = 361
First-degree relative(s) with colon cancer, *n* (%)	22 (6.5), *n* = 337
Presence of at least one extraintestinal manifestation, *n* (%)	205 (54.1)
At least one prior IBD-related intestinal resection, *n* (%)	201 (53.0)
Prior IBD-related intestinal resections per disease year, median (IQR)	0.029 (0.090)
Active cigarette smoking at first presentation, *n* (%)	110 (29.4), *n* = 374
History of any immunomodulator treatment, *n* (%)	347 (91.6)
Azathioprine, *n* (%)	284 (74.9)
Response, *n* (%)	172 (60.6)
Nonresponse, *n* (%)	20 (7.0)
Discontinuation of therapy for other reasons, *n* (%)	92 (32.4)
6-Mercaptopurine, *n* (%)	29 (7.7)
Response, *n* (%)	9 (31.0)
Nonresponse, *n* (%)	1 (3.4)
Discontinuation of therapy for other reasons, *n* (%)	19 (65.5)
Methotrexate, *n* (%)	81 (21.4)
Response, *n* (%)	46 (56.8)
Nonresponse, *n* (%)	7 (8.6)
Discontinuation of therapy for other reasons, *n* (%)	28 (34.6)
History of anti-TNFα treatment, *n* (%)	306 (80.7)
Infliximab, *n* (%)	157 (41.4)
Response, *n* (%)	115 (73.2)
Nonresponse, *n* (%)	14 (8.9)
Discontinuation of therapy for other reasons, *n* (%)	28 (17.8)
Adalimumab, *n* (%)	255 (67.3)
Response, *n* (%)	213 (83.5)
Nonresponse, *n* (%)	21 (8.2)
Discontinuation of therapy for other reasons, *n* (%)	21 (8.2)
Golimumab, *n* (%)	8 (2.1)
Response, *n* (%)	5 (62.5)
Nonresponse, *n* (%)	2 (25.0)
Discontinuation of therapy for other reasons, *n* (%)	1 (12.5)
History of anti-integrin treatment, *n* (%)	92 (24.3)
Response, *n* (%)	62 (67.4)
Nonresponse, *n* (%)	20 (21.7)
Discontinuation of therapy for other reasons, *n* (%)	10 (10.9)
History of anti-interleukin-12/23 treatment, *n* (%)	130 (34.3)
Response, *n* (%)	95 (73.1)
Nonresponse, *n* (%)	19 (14.6)
Discontinuation of therapy for other reasons, *n* (%)	16 (12.3)
Different prior IBD therapies from first diagnosis to baseline, *n*, median (IQR)	2.0 (3.0)
Number of different IBD therapies per disease year, median (IQR)	0.160 (0.230)
Prior exposure to therapies	
0 therapy, *n* (%)	32 (8.4)
1 therapy, *n* (%)	70 (18.5)
2 therapies, *n* (%)	98 (25.9)
3 therapies, *n* (%)	68 (17.9)
4 therapies, *n* (%)	47 (12.4)
5 therapies, *n* (%)	37 (9.8)
6 therapies, *n* (%)	21 (5.5)
7 therapies, *n* (%)	6 (1.6)

CD: Crohn’s disease; IBD: inflammatory bowel disease; IQR: interquartile range; SD: standard deviation; TNFα: tumor necrosis factor α; Discontinuation of therapy for other reasons includes: adverse events, infections, incompliance, diagnosis of cancer, death not related to therapy, poor vein status or reason for therapy discontinuation unknown.

**Table 3 genes-12-00866-t003:** Genotyping results.

Variable	CD	
Genotyping		RAF, %
NOD2 rs2066844, *n* (CC:CT:TT)	269:61:8; *n* = 338	11.3
NOD2 rs2066845, *n* (GG:GC:CC)	296:36:5; *n* = 338	6.8
NOD2 rs2066847, *n* (--:--C:CC)	277:55:6; *n* = 338	9.9
IL23R rs11209026, *n* (GG:AG:AA)	351:25:0; *n* = 376	3.3
PTPN2 rs2542151, *n* (TT:GT:GG)	260:101:12; *n* = 373	16.8
PTPN2 rs7234029, *n* (AA:AG:GG)	258:104:11; *n* = 373	16.9
ATG16L1 rs2241880, *n* (TT:CT:CC)	64:161:137; *n* = 362	60.1

ATG16L1: autophagy related 16 like 1; CD: Crohn’s disease; IL23R: interleukin-23 receptor; NOD2: nucleotide oligodimerization domain 2; PTPN2: protein tyrosine phosphatase non-receptor type 2; RAF: risk allele frequencies.

**Table 4 genes-12-00866-t004:** Baseline characteristics in 110 CD patients treated with anti-interleukin-12/23 according to PTPN2 rs7234029 polymorphism.

	PTPN2 rs7234029 (*n* = 110)		
Variable	Wild Type AA	Risk Type AG and GG	*p*-Value
*n* (%)	79 (71.8)	31 (28.2)	
Male, *n* (%)	40 (50.6)	16 (51.6)	0.926
Disease duration at baseline (years), mean ± SD	18.0 ± 11.4	18.2 ± 11.1	0.826
Age at diagnosis (years), mean ± SD	28.8 ± 13.0	30.6 ± 14.7	0.519
Montreal classification of CD:			
Age, *n* (A1:A2:A3)	8:55:16	5:19:7	0.617
Phenotype, *n* (L1:L2:L3)	29:7:43	9:4:18	0.635
L4, *n* (%)	8 (10.1)	2 (6.5)	0.722
Behavior, *n* (B1:B2:B3)	22:25:28, *n* = 75	8:6:17	0.208
Perianal disease, *n* (%)	23 (30.7), *n* = 75	15 (48.4)	0.084
First-degree relative(s) with IBD, *n* (%)	11 (14.5), *n* = 76	8 (25.8)	0.164
First-degree relative(s) with colon cancer, *n* (%)	7 (9.7), *n* = 72	3 (10.3), *n* = 29	1.000
Active smoking, *n* (%)	24 (32.9), *n* = 73	10 (34.5), *n* = 29	0.887
Presence of at least one extraintestinal manifestation, *n* (%)	46 (58.2)	19 (61.3)	0.769
At least one prior IBD-related intestinal resection, *n* (%)	48 (60.8)	20 (64.5)	0.715
Prior IBD-related intestinal resections per disease year, median (IQR)	0.043 ± 0.140	0.042 ± 0.090	0.637

CD: Crohn’s disease; IBD: inflammatory bowel disease; IQR: interquartile range; SD: standard deviation.

## Data Availability

Not applicable.

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
