# Peer review of "Gene Polymorphisms of NOD2, IL23R, PTPN2 and ATG16L1 in Patients with Crohn’s Disease: On the Way to Personalized Medicine?"

_genes, 2021, doi:10.3390/genes12060866_

Round 1
Reviewer 1 Report
Peter Hoffman, David Lamerz, Petra Hill, Marietta Kirchner, Annika Gauss
Gene Polymorphisms of NOD2, IL23R, PTPN2 and ATG16L1 in patients with inflammatory bowel diseases: on the way to personalized medicine?
This is an uncontrolled retrospective monocentric study utilising the IBD Registry, aimed to shed light on one of the most difficult questions within the Single Nucleotide Polymorphism era – can we associate an SNP risk allele with a clinical outcome, namely a response to therapy.
Study design:
The retrospective analysis was designed as a trawl of significant granularity of clinical data available at a single tertiary centre, looking for significance between those with CD /UC associated SNP risk alleles and those without. The SNPs the authors used are well-founded in the literature and already have some associations with clinical outcomes; these were referenced appropriately. The cohort of patients were those with moderate to severe disease. There are no power calculations present in the paper to determine whether the cohort size would identify true associations. There is no indication why the study lasted for 3 years and didn’t extend to provide a greater wealth of cohort. Small cohorts can limit conclusions of studies of clinical parameters with SNPs – and this appears to be the case here.
For the clinical data - there is no mention of the use of corticosteroids, which is a significant confounder when determining whether a patient has responded or not to therapy.
Why were 6 months used to determine whether the patient has responded to therapy, e.g. whether they are still on the drug? Response to therapy is denoted by clinical parameters, endoscopic and biochemical markers – the presence of a drug still being used is not a validated marker of whether the patient has responded to that therapy.
For the CD patients – being a tobacco smoker has the largest impact on response to therapy – it would be useful to know the percentage of smokers within each sub-cohorts – although the numbers are likely too small to look at multivariate analysis.
Materials and methods:
The primary and secondary study objectives have the same outcomes (line 99 and 102) – therapy outcomes. Please revise. The secondary outcomes appear to be baseline characteristics only.
There appears to a formatting error/sentence or word missing lines 104-105. ‘Informed consent to participate in the local IBD Biobank by donating a blood sample’ does not make sense.
Some clarity is required between using the word Biobank and Registry, please – they can mean very different things.
Results and analysis:
In the introduction, there was mention of the need for biomarkers and the intimation that SNPs could be used as a biomarker, which requires a clear association between a risk allele and a clinical parameter. A ‘risk allele’ is mentioned after every SNP rs code; which risk allele are you referring to? In all large SNP clinical association studies, a singular risk allele is identified. Due to the cohort size, the analysis was undertaken on heterozygotes and homozygotes combined. You need to be clear if you are going to do that for the protein-coding SNPs that the presence of any other allele other than the wildtype does not alter the amino acid between the non-wild types (e.g. they have the same effect) if you think that these changes have a functional effect. For the intronic SNPs such as rs7234029 –the functional effect is less clear without analysis looking for transcription factor binding sites, miRNA target sites etc., which is outside of the scope of this paper – however, the presence of a homozygous allele or heterozygous allele impacts on the functionality.
For the UC analysis – there was no statistical significance in therapy responses – the numbers are too small to comment even ‘loosely’. An option is to remove the UC segment as you aren’t providing any new information to the field and concentrate on the CD cohort – the CD cohort would be strengthened significantly by larger numbers.
Discussion:
The cohort is a very specific, niche, severe disease cohort, which does not reflect the large majority of IBD clinical cohorts of hospitals across Europe, but this is an appropriate cohort to answer the question concerning therapeutic response to biologics. However, the cohort subgroup numbers are too small to be able to undertake a multivariate analysis, which is what is needed to identify clinical parameter associated with SNP alleles that stand up robustly to further analysis in different cohorts.
There is a spelling mistake on line 277 and 278 – homozygous and heterozygous, please.
You mention the MAFs of the SNPs (line 287) – can you give them please (and how you calculated them, as you don’t have controls)
The rs7234029 risk allele (AG and GG) was found in 31 patients with CD who did not respond to ustekinumab. I agree strongly that larger cohorts are needed – the numbers are unfortunately too low for it to be identified as a biomarker currently. Hence, the statement that this is a strong association needs to be watered down.
A significant proportion of the discussion is based around ‘tendency' with low numbers and no statistical significance; care should be taken when having significantly large parts of the paper discussion based on non-scientifically supported outcomes.
In summary, this is an interesting paper that has the potential to add to the field; however, there are significant limitations and confounders which need to be acknowledged or have more discussion around them. Ideally, a larger cohort size is required.
Author Response
Study design:
The retrospective analysis was designed as a trawl of significant granularity of clinical data available at a single tertiary centre, looking for significance between those with CD /UC associated SNP risk alleles and those without. The SNPs the authors used are well-founded in the literature and already have some associations with clinical outcomes; these were referenced appropriately. The cohort of patients were those with moderate to severe disease. There are no power calculations present in the paper to determine whether the cohort size would identify true associations. There is no indication why the study lasted for 3 years and didn’t extend to provide a greater wealth of cohort. Small cohorts can limit conclusions of studies of clinical parameters with SNPs – and this appears to be the case here.
Response: Thank you for your valuable comment. Indeed, small cohorts limit the conclusion of a study. That is why we wished to have a larger cohort. However, our initial rough estimation of achievable patient numbers showed that only few more patients would have been able to be included in our study in the future. The reason for this is that the cohort of patients who visit our IBD outpatient clinic regularly is limited by our available personnel, which prompted us to make the decision to mainly see patients just for a second opinion around the time of the end of our study (excluding the possibility to evaluate their therapy responses).
Our study is a retrospective, exploratory analysis including all the data available at our treatment center. We present results of descriptive analyses which do not have to be interpreted in a confirmatory way. As power analysis is a key component for planning clinical trials, post hoc power analyses do not provide sensible results as they do not indicate true power for detecting statistical significance (Zhang Y, et al. General Psychiatry 2019;32:e100069. Doi:10.1136/gpsych-2019-100069).
For the clinical data - there is no mention of the use of corticosteroids, which is a significant confounder when determining whether a patient has responded or not to therapy.
Response: Thank you, this has also been a concern in other of our retrospective studies. However, we have unfortunately learned that due to the fact that many IBD patients take and taper steroids on their own, there is no possibility to reliably measure cumulative steroid doses. This is why – after thorough discussions within our study team – we finally decided on choosing a larger time frame than commonly used to assess therapy response in our study: based on our experience, the type of immunosuppressive therapy is usually changed in patients who are not able to taper off their steroids at the most until month 6 after the initiation of a new immunosuppressive medication. This is why over 6 months witch could not be tapered off. Therefore, we suggest that our study results should not be relevantly influenced by steroid comedication.s
Why were 6 months used to determine whether the patient has responded to therapy, e.g. whether they are still on the drug? Response to therapy is denoted by clinical parameters, endoscopic and biochemical markers – the presence of a drug still being used is not a validated marker of whether the patient has responded to that therapy.
Response (please refer also to the above response): We chose the end point of 6 months due to the retrospective character of the study. Endoscopic results as well as fecal calprotectin concentrations could not be included due to limited availability and also to the bias which would have been generated based on the observation that patients with a successful therapy would often not agree on performing colonoscopy or providing a stool sample.
For the CD patients – being a tobacco smoker has the largest impact on response to therapy – it would be useful to know the percentage of smokers within each sub-cohorts – although the numbers are likely too small to look at multivariate analysis.
Response: Due to the limited number of cases in our study, we were unfortunately not able to perform multivariate analysis. However, univariate analysis failed to identify tobacco smoking as a factor influencing our results (see Table 4). In the revised version of the manuscript, we include information on active tobacco smoking at the timepoint of therapy.
Materials and methods:
The primary and secondary study objectives have the same outcomes (line 99 and 102) – therapy outcomes. Please revise. The secondary outcomes appear to be baseline characteristics only.
Response: Thank you for the hint. In the revised version of the manuscript, secondary outcomes were changed to baseline characteristics.
There appears to be a formatting error/sentence or word missing lines 104-105. ‘Informed consent to participate in the local IBD Biobank by donating a blood sample’ does not make sense.
Response: The word „by“ was changed to „and“.
Some clarity is required between using the word Biobank and Registry, please – they can mean very different things.
Response: We apologize for the confusion of terms. In fact, the correct term is a biobank with the possibility to link the donated samples with clinical data that were collected in a structured way. We removed the term „registry“ from the revised version of the manuscript.
Results and analysis:
In the introduction, there was mention of the need for biomarkers and the intimation that SNPs could be used as a biomarker, which requires a clear association between a risk allele and a clinical parameter. A ‘risk allele’ is mentioned after every SNP rs code; which risk allele are you referring to? In all large SNP clinical association studies, a singular risk allele is identified. Due to the cohort size, the analysis was undertaken on heterozygotes and homozygotes combined. You need to be clear if you are going to do that for the protein-coding SNPs that the presence of any other allele other than the wildtype does not alter the amino acid between the non-wild types (e.g. they have the same effect) if you think that these changes have a functional effect. For the intronic SNPs such as rs7234029 –the functional effect is less clear without analysis looking for transcription factor binding sites, miRNA target sites etc., which is outside of the scope of this paper – however, the presence of a homozygous allele or heterozygous allele impacts on the functionality.
Response: Indeed, little is known on functional effects of rs7234029. We suggested the need for functional studies in the conclusion section of the revised manuscript (line366) and added this point in the limitations (line 357-359).
For the UC analysis – there was no statistical significance in therapy responses – the numbers are too small to comment even ‘loosely’. An option is to remove the UC segment as you aren’t providing any new information to the field and concentrate on the CD cohort – the CD cohort would be strengthened significantly by larger numbers.
Response: This is a good suggestion. Following your advice, we decided to exclude the UC cohort from the manuscript and to focus on the CD cohort with the relevant results. The whole manuscript was revised accordingly.
Discussion:
The cohort is a very specific, niche, severe disease cohort, which does not reflect the large majority of IBD clinical cohorts of hospitals across Europe, but this is an appropriate cohort to answer the question concerning therapeutic response to biologics. However, the cohort subgroup numbers are too small to be able to undertake a multivariate analysis, which is what is needed to identify clinical parameter associated with SNP alleles that stand up robustly to further analysis in different cohorts.
Response: In the end of the Discussion section, line 356 of the revised manuscript, we emphasize this point as a limitation of our study.
There is a spelling mistake on line 277 and 278 – homozygous and heterozygous, please.
Response: The spelling mistake was corrected.
You mention the MAFs of the SNPs (line 287) – can you give them please (and how you calculated them, as you don’t have controls)
Response: We are very sorry, we meant the risk allele frequency in our cohort.
The rs7234029 risk allele (AG and GG) was found in 31 patients with CD who did not respond to ustekinumab. I agree strongly that larger cohorts are needed – the numbers are unfortunately too low for it to be identified as a biomarker currently. Hence, the statement that this is a strong association needs to be watered down.
Response: „Strong association“ was changed to „association“.
A significant proportion of the discussion is based around ‘tendency' with low numbers and no statistical significance; care should be taken when having significantly large parts of the paper discussion based on non-scientifically supported outcomes.
Response: Thank you for mentioning, as we were feeling the same way about it. By removing the whole UC section, this problem could be mostly fixed.
Reviewer 2 Report
This study aimed to analyze the impact of some SNPs on Crohn’s disease and ulcerative colitis.
The paper is well written; however, some minor adjustments need to be done. For example, the line 38. The line 55-56, the authors mentioned several studies revealing associations between NOD2 gene mutations. However, there is only one study used as reference. One study is not “several studies”. Just add more references. The line 90; the way to inform the date varies from different countries. However, as this paper is written in English, it would be better to keep months, days, and years as a pattern. The issue observed in the line 38 is repeated in the line 145. I would just suggest the authors to include an illustration to summarize the discussion/conclusion of this study.
Author Response
Response to reviewer:
In lines 38 and 145, the changes were made according to your suggestions. Also, the date in line 90 was adapted.
In lines 55-56, a second reference was inserted.
We feel that the suggestion to summarize the discussion/conclusion in an illustration probably arises from a certain lack of clearness of these sections that we created by listing and discussing even tendencies of results, which was mainly the case in our UC cohort. Following the suggestion of your Co-reviewer, we decided to remove the complete UC cohort from the manuscript, as – mainly for small sample size – it did not add much to the paper except for confusing results. We think that removing this whole part, the discussion became clearer, so that we would prefer to not add an extra illustration.
Round 2
Reviewer 1 Report
Many thanks for the revision - I really like what you've done with it - thank you for all the responses.